# Effect of Selected Mechanical/Physical Pre-Treatments on *Chlorella vulgaris* Protein Solubility

Maria P. Spínola [1,2,†] , Mónica M. Costa [1,2,†] and José A. M. Prates [1,2,*]

1    CIISA—Centre for Interdisciplinary Research in Animal Health, Faculty of Veterinary Medicine, University of Lisbon, Av. da Universidade Técnica, 1300-477 Lisboa, Portugal; mariaspinola@fmv.ulisboa.pt (M.P.S.); monicacosta@fmv.ulisboa.pt (M.M.C.)
2    Associate Laboratory for Animal and Veterinary Sciences (AL4AnimalS), Faculty of Veterinary Medicine, University of Lisbon, Av. da Universidade Técnica, 1300-477 Lisboa, Portugal
\*    Correspondence: japrates@fmv.ulisboa.pt
†    These authors contributed equally to this work.

**Abstract:** *Chlorella vulgaris* has been recognized as an interesting alternative feeding source since it contains a good amount of high-quality protein. However, the presence of a recalcitrant cell wall strongly affects the nutrients' digestibility, bioaccessibility, and bioavailability. The present study aimed to determine the influence of different pre-treatments (bead milling, extrusion, freeze-drying, heating, microwave, and sonication) on *C. vulgaris*' protein solubility. For total protein content and solubility, the Bradford method and sodium dodecyl sulphate-polyacrylamide gel electrophoresis (SDS-PAGE) quantification were used, respectively, and protein degradation was assessed by SDS-PAGE through quantification of protein fractions (26 kDa, 32–40 kDa, 66–96 kDa, and others). The *o*-phthaldialdehyde assay was used for peptide formation. While there were no statistically significant differences for total soluble protein measurements in the supernatant fractions, the results showed an increase in larger proteins following bead milling and microwave pre-treatments, and sonication led to higher fractions of the remaining protein (mostly of low molecular weight). Nevertheless, extrusion significantly increased the release of peptides in the soluble fractions, and, considering industrial applicability, this method may be a better choice for improving *C. vulgaris* protein bioaccessibility in monogastric diets.

**Keywords:** microalga; *Chlorella vulgaris*; pre-treatment; protein; solubility

## 1. Introduction

Over the years, there has been growing interest in using microalgae to replace common protein sources, partially or completely, such as soybeans or beans [1]. Various species of microalgae are consumed worldwide, with *Arthrospira* sp., *Chlorella* sp., *Isochrysis* sp., *Porphydium* sp., and *Schizochytrium* sp. being among the most prevalent. In many Asian countries, microalgae are already used as feed supplements and according to Hashemian, et al. [1], Souza, et al. [2] and Saadaoui, et al. [3], about 30% of total microalgae production is used for animal feeding, particularly for aquaculture fish, livestock animals, and pets [4–7]. The inclusion of whole microalgae in livestock diets, at incorporation levels between 5.5% and 10% for swine and poultry, respectively, and 20% for lambs, was reported to have no adverse effects on animal growth, while microalga supplementation was shown to improve animal performance and meat quality due to algal fatty acid profile and bioactive compounds [8]. For instance, Alfaia, et al. [9] tested the inclusion of 10% *Chlorella vulgaris*, either alone or in combination with enzymes, for 14 days in broilers' diet, and found an improvement in breast fatty acid profile. Additionally, Roques, et al. [6] showed that feeding broilers with 0.8% *C. vulgaris* for 35 days improved their performance. Coelho, et al. [10] and Martins, et al. [11] observed that the inclusion of 5% *C. vulgaris* for 14 days in finishing pigs' and post-weaned piglets' diets, respectively, increased antioxidants, pigments, and

n-3 polyunsaturated fatty acids in meat. Overall, the inclusion of microalgae in animal diets offers several benefits for animal health, such as improved immune response, disease resistance, intestinal health, and stimulation of prebiotic microbiota [12]. Moreover, microalgae are considered a sustainable and rapidly growing alternative feed source [13], providing a good amount of biomass containing high levels of protein, energy [14], and essential amino acids, such as methionine and lysine, which are limiting amino acids in farming animal diets [8].

*Chlorella*, as a genus of green microalgae (*Chlorophyceae*), has an outer layer called algaenan, which is also known as sporopollenin, in the structure of the cell wall that influences the susceptibility to disruption methods [15,16]. However, the cell wall structure can change according to growth stage and culture conditions [15,17]. In general, *Chlorella* is one of the genera with the lowest algaenan content in the cell wall among *Chlorophyceae*, although it presents a fibrillary cell wall consisting of two main layers or glycoprotein structures composed of glucose, mannose, glucosamine, algaenan, and/or β-galactofuranan [15,18]. In addition, Canelli, et al. [18] concluded that the cell wall polysaccharide profile is similar between growth and exponential phases of microalga development, with a predominance of rhamnose and galactose.

*Chlorella vulgaris* is normally rich in proteins, lipids, carbohydrates, vitamins, pigments, and minerals, depending on factors such as growth conditions, species, and genus [2]. According to Baudelet, et al. [17] and Safi, et al. [19], *C. vulgaris* in the initial phase has a fragile and thin cell wall, and, as the microalga grows, the cell wall becomes thicker and more rigid. The presence of a recalcitrant cell wall in *C. vulgaris* makes it difficult for nutrients to be bioaccessible and digestible, especially for monogastric animals. As a microalga rich in protein, this nutrient digestibility is important to bear in mind and several factors can influence it, such as the composition of the cell wall and its rigidity or the strain [20].

The potential of *Chlorella sorokiniana* and *C. vulgaris* proteins as bioactive peptides and the protein profiles of these microalgae under different growth conditions have been investigated. Tejano, et al. [21] identified eight proteins from *C. sorokiniana* that have the potential for becoming bioactive peptides, including Fe-superoxide dismutase chloroplast rubisco activase, heat shock protein, and phosphoglycerate kinase. These proteins had molecular weights ranging from 109 to 7.82 kDa. Khairy, et al. [22] compared the protein profiles of *C. vulgaris* grown autotrophically and heterotrophically and found no significant differences between the two growth conditions. However, the autotrophic condition had higher intensity of bands with 75 and 39 kDa. In another study, Piasecka and Baier [23] investigated the influence of three cultivation modes (autotrophic, photoheterotrophic, and mixotrophic) on the protein profile of *C. vulgaris*. Their analysis using SDS-PAGE showed that the photoheterotrophic condition produced two protein fractions, one between 49 and 77 kDa, located in the cytoskeleton, and another fraction between 70.9 and 80.7 kDa, located in the chloroplast in response to stress. Overall, these studies provide valuable information on the protein profiles and potential bioactive peptides of *Chlorella* species under different growth conditions. According to Van De Walle, et al. [20], protein isolation can improve its accessibility and digestibility, although the cost/effectiveness of this procedure might be an issue.

Efficient extraction of intracellular components, such as proteins and lipids, depends on how great the disruption of microalgae cell walls is. Therefore, applying pre-treatments, such as mechanical, physical, or enzymatic methods, can be a viable solution to this issue [15,20,24,25]. Bead milling (BM) is a widely used technique that directly affects the cells by applying high-speed spinning with fine beads, and the effectiveness of disruption is influenced by the number of beads used and their diameter [26,27]. Nevertheless, the percentage of microalga present in suspension and the characteristics of the shaker (speed and type) also influences the success of cell wall disintegration [28]. Although BM is a simple method, a study by Zheng, et al. [29] showed that it did not improve lipid concentration in *C. vulgaris*. Extrusion (ET) is another pre-treatment that involves heat,

compression, mixing, and shear forces to disrupt and modify the biomass before passing through an extruder [30]. Wang, et al. [31] demonstrated that ET is a good alternative for disrupting the cell wall of *Nannochloropsis oceanica* and increasing the recovery of lipids. Heating (HT) using the autoclaving method is a commonly used technique due to its lower costs. However, high temperatures can increase the possibility of lipids and other compounds forming a complex that hinders solvent extraction [27,32]. Freeze-drying (FD) is used for both drying and disrupting cells as it removes the water present in frozen microalgae [33,34]. Unterlander, et al. [35] demonstrated that FD improved the extraction of soluble proteins and active enzymes from *C. vulgaris* biomass. Microwave (MW) radiation uses thermal effects to heat polar solvents in contact with solid samples, and the combination of temperature and pressure provokes the liberation of bioactive compounds from the cells by friction [34,36]. MW can be applied on a large scale. Sonication (SO) is another pre-treatment that induces cell wall disruption through cavitation waves generated by ultrasounds [26,32]. Although Zheng, et al. [29] found that SO did not improve lipid concentration in *C. vulgaris*, Piasecka, et al. [32] demonstrated that SO was effective and improved lipid release compared to non-pre-treated microalga.

The objective of this study was to assess and compare the effectiveness of six different pre-treatments (i.e., bead milling, extrusion, freeze-drying, heating, microwave radiation, and sonication) to disrupt the cell wall of *C. vulgaris* and improve the extraction of soluble proteins. Therefore, we analyzed total protein and peptide content and solubility and specific protein fractions from this microalga. Ultimately, we intended to identify the most effective pre-treatment methods.

## 2. Materials and Methods

### 2.1. Microalga Pre-Treatments

*Chlorella vulgaris* dried powder was obtained from Allmicroalgae company (Pataias, Portugal). In Table 1 is presented the chemical composition of non-treated *C. vulgaris*, provided by Allmicroalgae company. Then, it was submitted to six different pre-treatments with specific protocols, as described in Costa, et al. [37] and Spínola, et al. [38].

**Table 1.** Chemical composition of non-treated *Chlorella vulgaris* (expressed as % dry matter, except energy).

| Nutritional Composition | |
|---|---|
| Energy (MJ/kg) | 15.5 |
| Crude protein | 31.3 |
| Ash | 4.48 |
| Crude carbohydrates | 36.3 |
| Crude fiber | 19.6 |
| Crude fat | 8.44 |
| Pigment composition | |
| Chlorophyll | 0.60 |
| Total carotenoids | 0.16 |

Concisely, bead milling pre-treatment consisted of the homogenization, at 2000 rpm for 30 min in a shaker (Multi Reax Heidolph Instruments, Schwabach, Germany), of 20 mg microalga/mL of $1 \times$ Phosphate Buffered Saline (PBS) solution (BioWhittaker, Verviers, Belgium) using 0.5 mm diameter zirconium beads (one per mL). The extrusion method for *C. vulgaris* was also performed by Sparos company (Olhão, Portugal) but the temperature of the last barrel was 114 °C instead of 118 °C. The rest of the variables, such as pressure (34 bars), water addition (for 3 to 7 s, 340 mL/min), and drying (120 °C for 8 to 10 min) were applied as reported by Costa, et al. [37] and Spínola, et al. [38]. The freeze-drying method consisted of freezing the microalga at −80 °C for 24 h and then freeze-drying it (Labogene, CoolSafe, Frilabo, Milheirós, Portugal) for another 24 h. In the heating technique, the

microalga was dried (Melag, Geneststraße, Berlin, Germany) at 70 °C for 30 min. For the microwave (Whirlpool, Household Microwave Oven, MI, USA) method, a microalga resuspension was kept in keep warm mode until it boiled. For sonication (Bandelin ultrasonic homogenizer, Heinrichstraße, Berlin, Germany), a microalga resuspension was submitted to 7 cycles at 70% power for 15 min with manual agitation in the meantime.

The chemical composition was determined using routine and common methods [39]. Briefly, for dry matter, the sample was dried at 105 °C to a constant weight. Crude protein determination by nitrogen content was obtained by the Kjeldahl method. Ash content was determined after burning the sample at 525 °C. Crude fat value was determined by Soxhlet extraction with petroleum. Gross energy and crude carbohydrates were obtained by common and regular methods, as described by Costa, et al. [37] and Spínola, et al. [38]. Pigments, such as chlorophyll and total carotenoids, were extracted by high-performance liquid chromatography, as described by Ritchie [40].

### 2.2. Incubation of C. vulgaris after Pre-Treatments

*C. vulgaris* resuspension at 20 mg/mL in 1 × Phosphate Buffered Saline (PBS) solution (BioWhittaker, Verviers, Belgium) was incubated overnight with each pre-treatment ($n = 5$) in an orbital shaker (Sanyo MIR-220RU Refrigerated, Shaking Incubator, Osaka, Japan), as described in Costa, et al. [37] and Spínola, et al. [38].

### 2.3. Total Protein Content by Bradford Method

To quantify solubilized total protein by spectrophotometry in pellet and supernatant, the Bradford method was used according to Costa, et al. [37] and Spínola, et al. [38], with the exception that the sample was not previously diluted.

### 2.4. Protein Fraction Quantification and Solubility of C. vulgaris Proteins by SDS-PAGE

Microalga proteins were separated by 12% SDS-PAGE [21], following conditions described in Costa, et al. [37] and Spínola, et al. [38] but with some modifications. The mixture of each sample and sample buffer was added at 8 μL into gel wells, while the low-molecular-weight (LMW) protein marker (18.5, 26, 32, 40, 48, 90, and 96 kDa protein bands) was added at 5 μL (9.00 μg of protein) (NZYTech, Lisbon, Portugal).

Using the Image J software (version 1.53s) (NIH, Bethesda, MA, USA), the three most prominent protein fractions (66–96 kDa, protein fraction 1, F1; 32–40 kDa, protein fraction 2, F2; and 26 kDa, protein fraction 3, F3), other proteins, and total protein were quantified. The band of 40.0 kDa of the LMW marker (1.95 μg of protein) was used as an external standard for this determination. The proportion of these fractions (F1, F2, F3, and others) was calculated relative to total protein.

### 2.5. Total Peptides by O-Phthaldialdehyde Assay

The *o*-phthaldialdehyde (OPA) spectrophotometric assay was carried out as described in Costa, et al. [37] and Spínola, et al. [38].

### 2.6. Statistical Analysis

To investigate the data, general Linear Models of SAS (SAS Institute Inc., Cary, NC, USA) were used. The analysis of variance (ANOVA) and Tukey–Kramer method (PDIFF option) were performed for multiple comparisons of adjusted least square means. Levene's test was used for the homogeneity of variances. Values were considered significant when $p < 0.05$.

## 3. Results

Table 2 reflects the impact of the selected pre-treatments on the release and degradation of *C. vulgaris* protein in the supernatant fraction.

**Table 2.** Impact of pre-treatments on *Chlorella vulgaris* biomass proteins' solubility in the supernatant fraction ($n = 5$) (values are presented as mean ± standard deviation).

| Item | Pre-Treatments [1] | | | | | | | *p*-Value |
|---|---|---|---|---|---|---|---|---|
| | **NoP** | **BM** | **ET** | **FD** | **HT** | **MW** | **SO** | |
| Total protein (mg/mL) | | | | | | | | |
| Bradford method | 0.08 ± 0.042 | 0.12 ± 0.105 | 0.06 ± 0.030 | 0.09 ± 0.049 | 0.06 ± 0.029 | 0.10 ± 0.116 | 0.09 ± 0.005 | 0.698 |
| SDS-PAGE gel | 6.66 ± 0.162 | 7.43 ± 0.416 | 6.83 ± 0.308 | 6.73 ± 0.372 | 6.79 ± 0.230 | 7.42 ± 0.526 | 7.33 ± 0.661 | 0.081 |
| Proteins (mg/mL) in 12% SDS-PAGE gel | | | | | | | | |
| Proteins 66–96 kDa | 1.28 ± 0.115 [ab] | 1.50 ± 0.058 [a] | 1.26 ± 0.254 [ab] | 1.24 ± 0.026 [b] | 1.19 ± 0.095 [b] | 1.50 ± 0.081 [a] | 1.15 ± 0.061 [b] | <0.001 |
| Proteins 32–40 kDa | 1.26 ± 0.075 [b] | 1.51 ± 0.058 [a] | 1.27 ± 0.185 [b] | 1.18 ± 0.092 [b] | 1.18 ± 0.095 [b] | 1.48 ± 0.105 [a] | 1.17 ± 0.050 [b] | <0.001 |
| Protein 26 kDa | 1.39 ± 0.058 [ab] | 1.50 ± 0.058 [a] | 1.28 ± 0.152 [abc] | 1.32 ± 0.225 [abc] | 1.20 ± 0.046 [bc] | 1.50 ± 0.079 [a] | 1.13 ± 0.064 [c] | <0.001 |
| Other proteins | 2.73 ± 0.277 [b] | 2.93 ± 0.259 [b] | 3.02 ± 0.378 [b] | 2.99 ± 0.097 [b] | 3.23 ± 0.046 [b] | 2.94 ± 0.264 [b] | 3.88 ± 0.585 [a] | <0.001 |
| Proteins (% total protein) in 12% SDS-PAGE gel | | | | | | | | |
| Proteins 66–96 kDa | 19.2 ± 2.12 [a] | 20.2 ± 0.76 [a] | 18.4 ± 3.03 [ab] | 18.5 ± 0.74 [ab] | 17.5 ± 0.87 [ab] | 20.2 ± 0.43 [a] | 15.8 ± 1.66 [b] | 0.002 |
| Proteins 32–40 kDa | 19.0 ± 1.51 [ab] | 20.3 ± 0.40 [a] | 18.5 ± 2.11 [ab] | 17.5 ± 0.44 [bc] | 17.4 ± 0.82 [bc] | 20.0 ± 0.08 [a] | 16.0 ± 1.37 [c] | <0.001 |
| Protein 26 kDa | 20.9 ± 0.40 [a] | 20.2 ± 0.38 [a] | 18.8 ± 1.53 [ab] | 19.6 ± 2.37 [ab] | 17.6 ± 0.09 [bc] | 20.2 ± 0.44 [a] | 15.9 ± 1.11 [c] | <0.001 |
| Other proteins | 40.9 ± 3.31 [bc] | 39.4 ± 1.34 [c] | 44.3 ± 6.58 [bc] | 44.5 ± 2.26 [bc] | 47.6 ± 1.71 [ab] | 39.6 ± 0.79 [c] | 52.3 ± 4.12 [a] | <0.001 |
| Proteins (PTRAT/PCON) in 12% SDS-PAGE gel | | | | | | | | |
| Total protein | nd | 1.12 ± 0.045 | 1.03 ± 0.067 | 1.01 ± 0.037 | 1.02 ± 0.057 | 1.11 ± 0.059 | 1.10 ± 0.078 | 0.083 |
| Proteins 66–96 kDa | nd | 1.18 ± 0.142 [a] | 0.98 ± 0.116 [ab] | 0.98 ± 0.106 [ab] | 0.93 ± 0.048 [b] | 1.19 ± 0.163 [a] | 0.91 ± 0.082 [b] | 0.001 |
| Proteins 32–40 kDa | nd | 1.20 ± 0.114 [a] | 1.00 ± 0.093 [ab] | 0.94 ± 0.125 [b] | 0.93 ± 0.034 [b] | 1.18 ± 0.148 [a] | 0.93 ± 0.068 [b] | <0.001 |
| Protein 26 kDa | nd | 1.07 ± 0.024 [a] | 0.93 ± 0.147 [ab] | 0.95 ± 0.130 [ab] | 0.86 ± 0.067 [b] | 1.08 ± 0.028 [a] | 0.81 ± 0.017 [b] | <0.001 |
| Other proteins | nd | 1.08 ± 0.027 [b] | 1.11 ± 0.042 [b] | 1.10 ± 0.112 [b] | 1.19 ± 0.121 [b] | 1.08 ± 0.033 [b] | 1.42 ± 0.081 [a] | <0.001 |
| Total peptides (µg/mL) | | | | | | | | |
| *o*-phthaldialdehyde assay | 13.1 ± 2.94 [b] | 16.8 ± 5.06 [b] | 32.4 ± 5.34 [a] | 11.2 ± 1.94 [b] | 12.4 ± 3.31 [b] | 14.2 ± 5.43 [b] | 19.1 ± 2.79 [b] | <0.001 |

[1] No pre-treatment (NoP); Bead milling (BM); Extrusion (ET); Freeze-drying (FD); Heating (HT); Microwave (MW); Sonication (SO). SDS-PAGE, sodium dodecyl sulphate-polyacrylamide gel electrophoresis; PTRAT, protein obtained with pre-treatments; PCON, protein obtained with control. Values with significant ($p < 0.05$) increases compared to the NoP control are highlighted in bold. [a,b,c]: In the same line, different letters indicate statistically significant differences. nd, not detected.

Total protein content, assessed by the Bradford method, was not significantly affected by any pre-treatment. However, total protein solubility, determined by SDS-PAGE, tended to be higher ($p = 0.081$) with bead milling, microwave, and sonication. Regarding protein quantification in the gel, there was a highly significant ($p < 0.001$) effect of pre-treatments for every fraction, but only bead milling and microwave had significant differences in comparison with control, increasing the protein fraction with 32- 40 kDa. Sonication pre-treatment also significantly affected protein quantification in the gel. It decreased the protein fraction of 26 kDa and increased the fraction of other proteins. Only sonication significantly affected all the percentages of protein fractions, with a decrease in protein fractions with 26 kDa, 32–40 kDa, and 66–96 kDa, whereas it provoked an increase in other proteins fraction. The percentage of protein fraction with 26 kDa significantly decreased ($p < 0.001$) with heating pre-treatment. There was a significant increase ($p < 0.001$) in the proportion of protein fractions (protein with pre-treatments/protein with control; PTRAT/PCON) with 66–96 kDa, 30–40 kDa, and 26 kDa when bead milling and microwave were applied, relative to control. A significant increase ($p < 0.001$) in the fraction of other proteins with sonication pre-treatment was also observed. The extrusion was the only treatment that showed a significant increase in peptide formation quantified by OPA assay ($p < 0.001$).

Figure 1 shows representative images of SDS-PAGE gels concerning the effect of pre-treatments on *C. vulgaris* proteins, in supernatant fraction. The prominent lane between 66–96 kDa is clearly visible, although the bands for each pre-treatment cannot be distinguished between them in the gels. The poor visualization of protein bands in *C. vulgaris* gels is probably a consequence of a difficult cell wall disruption.

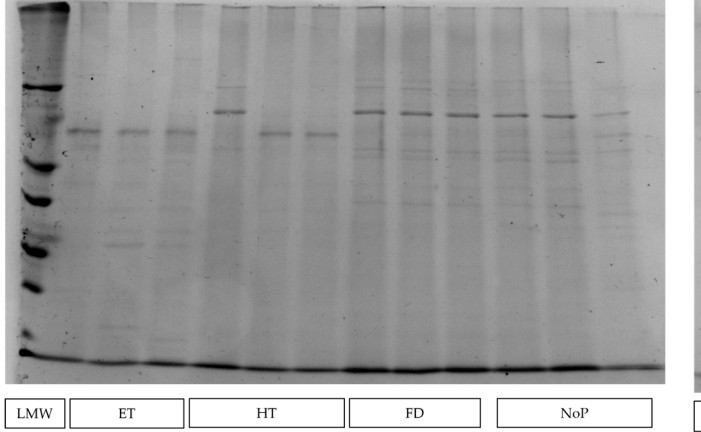 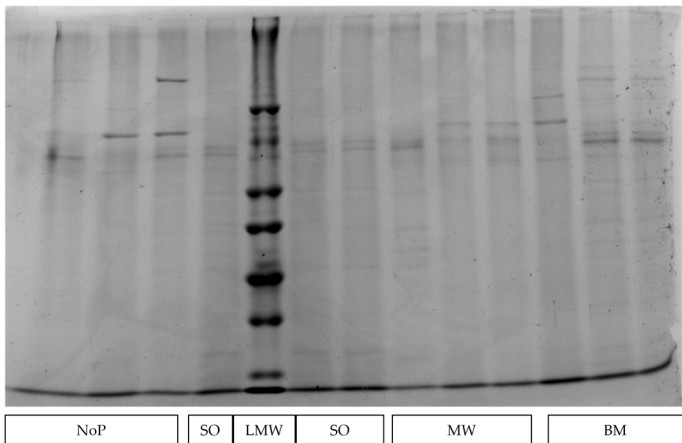

| LMW | ET | HT | FD | NoP | | NoP | SO | LMW | SO | MW | BM |

**Figure 1.** Representative images of 12% SDS-PAGE gels of the supernatant fraction showing the effect of pre-treatments on *Chlorella vulgaris* protein amount and solubility after 16 h of incubation ($n = 3$); LMW, Low-Molecular-Weight protein marker (18.5 to 96 kDa); No pre-treatment (NoP); Bead milling (BM); Extrusion (ET); Freeze-drying (FD); Heating (HT); Microwave (MW); Sonication (SO).

The influence of pre-treatments on the degradation of *C. vulgaris* protein biomass in the pellet fraction is shown in Table 3. In addition, the SDS-PAGE gels representative of the impact of these pre-treatments on *C. vulgaris* proteins are presented in Figure 2. The only effect on the total protein was a significant increase ($p < 0.001$) in that quantified by Bradford with bead milling pre-treatment. Considering protein quantification in the gel, protein fraction with 66 to 96 kDa had a significant ($p = 0.003$) increase with extrusion and heating methods. In addition, protein fraction with 32 to 40 kDa and other proteins were significantly affected, $p = 0.011$ and $p = 0.002$, respectively, by pre-treatments, but without significant differences when compared to the control. A similar result was found for the percentage of protein fractions with 66 to 96 kDa ($p = 0.007$) and 32 to 40 kDa ($p = 0.007$) and other proteins ($p = 0.043$). There was a significant decrease in the proportion of protein fraction (protein with pre-treatments/protein with control; PTRAT/PCON) with 32 to

40 kDa ($p$ = 0.008) relative to control with bead milling and microwave pre-treatments, whereas heating significantly increased this protein fraction. In addition, heating significantly ($p$ = 0.003) decreased the proportion of other proteins, while microwave increased it. Although the effect of treatments on the proportion of protein fraction with 26 kDa was significant ($p$ = 0.044), there were no significant differences between pre-treatments.

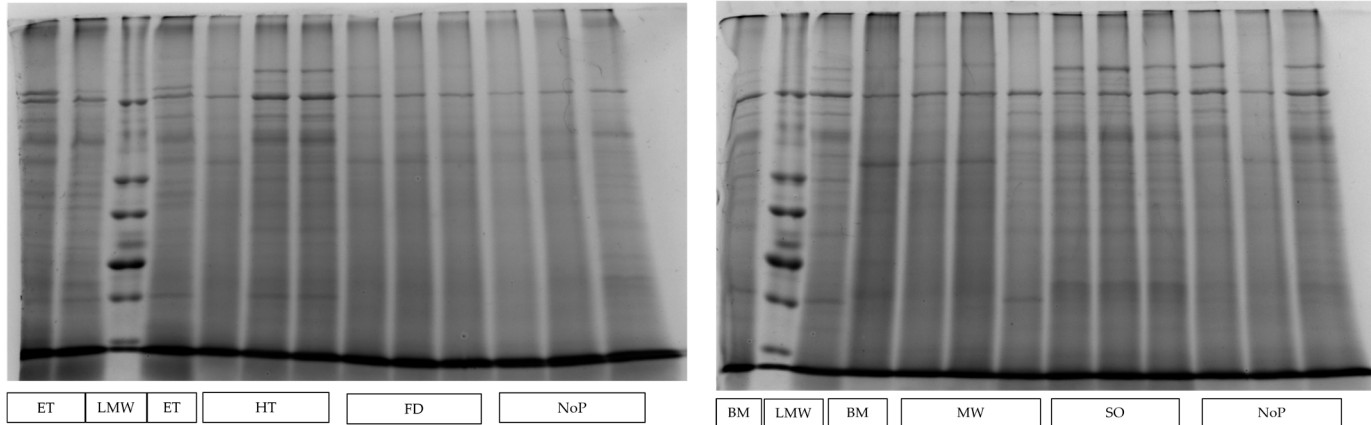

**Figure 2.** Representative images of 12% SDS-PAGE gels of the pellet fraction showing the effect of pre-treatments on *Chlorella vulgaris* protein amount and solubility after 16 h of incubation ($n$ = 3); LMW, Low-Molecular-Weight protein marker (18.5 to 96 kDa); No pre-treatment (NoP); Bead milling (BM); Extrusion (ET); Freeze-drying (FD); Heating (HT); Microwave (MW); Sonication (SO).

Figure 2 shows unclear differences between the lanes of each pre-treatment in the pellet fraction, in concordance with the supernatant fraction.

Table 3. Impact of pre-treatments on *Chlorella vulgaris* biomass proteins' solubility in the pellet fraction ($n$ = 5) (values are presented as mean ± standard deviation).

| Item | Pre-Treatments [1] | | | | | | | *p*-Value |
|---|---|---|---|---|---|---|---|---|
| | **NoP** | **BM** | **ET** | **FD** | **HT** | **MW** | **SO** | |
| Total protein (mg/mL) | | | | | | | | |
| Bradford method | 0.38 ± 0.205 [b] | 1.64 ± 0.129 [a] | 0.75 ± 0.365 [b] | 0.29 ± 0.140 [b] | 0.38 ± 0.199 [b] | 0.75 ± 0.197 [b] | 1.10 ± 0.956 [ab] | <0.001 |
| SDS-PAGE gel | 8.22 ± 0.319 | 8.88 ± 1.037 | 9.44 ± 1.194 | 8.42 ± 0.390 | 8.59 ± 0.225 | 9.20 ± 1.303 | 8.97 ± 0.285 | 0.235 |
| Proteins (mg/mL) in 12% SDS-PAGE gel | | | | | | | | |
| Proteins 66–96 kDa | 1.33 ± 0.212 [b] | 1.30 ± 0.198 [b] | 1.68 ± 0.030 [a] | 1.52 ± 0.083 [ab] | 1.65 ± 0.064 [a] | 1.39 ± 0.187 [ab] | 1.46 ± 0.198 [ab] | 0.003 |
| Proteins 32–40 kDa | 1.53 ± 0.158 [ab] | 1.33 ± 0.263 [b] | 1.75 ± 0.090 [ab] | 1.65 ± 0.093 [ab] | 1.76 ± 0.037 [a] | 1.35 ± 0.447 [ab] | 1.66 ± 0.080 [ab] | 0.011 |
| Protein 26 kDa | 1.68 ± 0.170 | 1.53 ± 0.314 | 1.78 ± 0.037 | 1.74 ± 0.094 | 1.82 ± 0.067 | 1.60 ± 0.359 | 1.86 ± 0.059 | 0.133 |
| Other proteins | 3.67 ± 0.461 [ab] | 4.72 ± 0.754 [a] | 4.24 ± 1.128 [ab] | 3.51 ± 0.563 [b] | 3.36 ± 0.184 [b] | 4.85 ± 0.332 [a] | 3.98 ± 0.089 [ab] | 0.002 |
| Proteins (% total protein) in 12% SDS-PAGE gel | | | | | | | | |
| Proteins 66–96 kDa | 16.2 ± 2.32 [ab] | 14.7 ± 2.85 [b] | 18.0 ± 2.11 [ab] | 18.0 ± 1.60 [ab] | 19.2 ± 0.75 [a] | 15.2 ± 0.68 [b] | 16.3 ± 1.83 [ab] | 0.007 |
| Proteins 32–40 kDa | 18.6 ± 1.61 [abc] | 16.8 ± 1.73 [c] | 18.6 ± 1.37 [abc] | 19.7 ± 1.77 [ab] | 20.5 ± 0.63 [a] | 17.7 ± 1.64 [bc] | 18.5 ± 0.37 [abc] | 0.007 |
| Protein 26 kDa | 20.5 ± 1.80 | 19.8 ± 1.78 | 19.1 ± 2.62 | 20.7 ± 1.64 | 21.2 ± 0.69 | 19.8 ± 2.01 | 20.8 ± 0.83 | 0.563 |
| Other proteins | 44.7 ± 5.70 [ab] | 48.6 ± 5.37 [a] | 44.3 ± 6.07 [ab] | 41.6 ± 4.98 [ab] | 39.1 ± 1.31 [b] | 47.3 ± 4.18 [ab] | 44.4 ± 1.37 [ab] | 0.043 |
| Proteins (PTRAT/PCON) in 12% SDS-PAGE gel | | | | | | | | |
| Total protein | nd | 1.08 ± 0.046 | 1.15 ± 0.046 | 1.03 ± 0.046 | 1.05 ± 0.046 | 1.12 ± 0.046 | 1.09 ± 0.046 | 0.425 |
| Proteins 66–96 kDa | nd | 0.99 ± 0.076 | 1.29 ± 0.076 | 1.16 ± 0.076 | 1.26 ± 0.076 | 1.05 ± 0.076 | 1.10 ± 0.076 | 0.079 |
| Proteins 32–40 kDa | nd | 0.87 ± 0.067 [b] | 1.15 ± 0.067 [ab] | 1.09 ± 0.067 [ab] | 1.16 ± 0.067 [a] | 0.87 ± 0.067 [b] | 1.09 ± 0.067 [ab] | 0.008 |
| Protein 26 kDa | nd | 0.91 ± 0.051 | 1.07 ± 0.051 | 1.04 ± 0.051 | 1.09 ± 0.051 | 0.94 ± 0.051 | 1.12 ± 0.051 | 0.044 |
| Other proteins | nd | 1.29 ± 0.076 [ab] | 1.15 ± 0.076 [abc] | 0.96 ± 0.076 [bc] | 0.92 ± 0.076 [c] | 1.34 ± 0.076 [a] | 1.10 ± 0.076 [abc] | 0.003 |

[1] No pre-treatment (NoP); Bead milling (BM); Extrusion (ET); Freeze-drying (FD); Heating (HT); Microwave (MW); Sonication (SO). SDS-PAGE, sodium dodecyl sulphate-polyacrylamide gel electrophoresis; PTRAT, protein obtained with pre-treatments; PCON, protein obtained with control. [a,b,c]: In the same line, different letters indicate statistically significant differences. nd, not detected.

## 4. Discussion

The results of this study suggest that mechanical/physical pre-treatments, such as bead milling and sonication, have an impact on the solubility and relative proportion of proteins in *C. vulgaris*. Specifically, sonication was found to decrease the solubility of *C. vulgaris* proteins, leading to significant decreases in protein fractions with molecular weights of 66 to 96 kDa, 32 to 40 kDa, and 26 kDa. In addition, bead milling, and microwave were found to be more effective in increasing the proportion of protein fractions with 66 to 96 kDa, 32 to 40 kDa, and 26 kDa compared to the control. Both pre-treatments increased the amount of proteins with 32 to 40 kDa released to the supernatant, which suggests their ability to disrupt microalgal cell walls, although these results are not presented in the gel due to a low protein extraction yield. Indeed, the pre-treatments had no significant effect on the concentration of total protein quantified either by the Bradford method or SDS-PAGE gel analysis.

The efficiency of bead milling in increasing the solubility of the protein fraction with 32 to 40 kDa is consistent with previous reports [41–43]. Alavijeh, et al. [41] demonstrated that bead milling with 0.2 mm beads for half a minute could extract 10% of soluble proteins from *C. vulgaris*. However, the authors reported an increase in a chloroplast-associated protein (rubisco), which has subunits of around 56 and 14 kDa [44,45]. Therefore, in the present study, the protein fraction released with bead milling probably did not correspond to rubisco, and other pigment-related proteins, particularly with molecular weight between 7.80 and 46.0 kDa [21], might have been released instead. The efficiency of bead milling for protein extraction was also reported in other studies, such as Günerken, et al. [42] and Kulkarni and Nikolov [43], with extraction yields up to 76%, even though the formation of emulsions during bead milling can complicate protein extraction and, thus, the processing of bioactive compounds [46]. The positive effects of bead milling on increasing *C. vulgaris* protein digestibility and bioavailability were previously reported in in vivo trials. Neumann, et al. [47] tested the effect of bead-milling-treated *C. vulgaris* in mice diets and observed improved protein bioavailability and quality parameters compared to the control diet, even at incorporation levels of up to 25%. Similarly, Batista, et al. [48] using a vibratory mill instead of a bead milling method for cell wall disruption in *C. vulgaris* in diets observed that this technique improved *C. vulgaris* protein's apparent digestibility coefficient for European seabass by 1.0% (91.6% in non-treated to 92.6% in vibratory milled microalga). These findings highlight the potential of bead milling as a method for extracting high-quality proteins from *C. vulgaris* for various applications.

Sonication reduced the solubility of the main protein fractions, with a significant decrease observed in protein fractions with 66 to 96 kDa, 32 to 40 kDa, and 26 kDa, although it increased the solubility of other minor protein fractions. The propagation of ultrasound waves in the medium generates high local pressure and temperature, which can cause denaturation and aggregation of microalga proteins with a consequent decrease in their solubility [49,50]. Previous studies reported that optimized sonication conditions could improve protein yield and separation efficiency in *C. vulgaris*, although the effect of this method on algal protein solubility was not assessed. Specifically, Chia, et al. [51] observed an increase in protein yield from 25.2% to 40.0% and separation efficiency from 49.8% to 52.3% with a sonication-assisted triphasic partitioning process. The authors suggested that this pre-treatment could be scaled up for industrial protein extraction and may also enhance the extraction of other bioactive compounds. Indeed, Gille, et al. [52] found that the bioaccessibility of lutein and β-carotene improved up to 18% and 12.5%, respectively, with this treatment. Moreover, Weber, et al. [53] described that the use of sonication for 30 min could release up to 17% of proteins and 9% of sugars after disruption of the *C. vulgaris* cell wall. Similarly, Janczyk, et al. [54] evaluated the effect of ultrasonication on apparent crude protein digestibility (ADCP) in rats and found that this pre-treatment improved ADCP by 9.8% (from 46.9% in spray-dried to 56.7% in ultrasonicated *C. vulgaris*).

Microwave treatment had a positive effect on the relative proportions of the main protein fractions in the supernatant and also caused an increase in the concentration of

protein fractions with 32 to 40 kDa. Other authors showed the benefits of using this method to increase the protein extraction yield of *C. vulgaris*. For example, Chew, et al. [36] combined microwave treatment with a three-phase partitioning technique and observed a 2.54-fold increase in protein yield. Moreover, an optimized microwave method, which involved adjusting the duty cycle and frequency, improved protein yield from 24.9% to 63.2% and separation efficiency from 46.8% to 67.2%. Therefore, the microwave treatment successfully disrupted the microalga cell wall and released bioactive compounds such as proteins. For both sonication and microwave treatments, it is important to consider some parameters, such as power and frequency, irradiation time, and duty cycle. These parameters can affect protein denaturation, the formation of undesirable compounds, disruption of the cell wall, and the overall success of the method [36,51].

In the present study, extrusion was not effective in extracting soluble proteins from *C. vulgaris*, but it increased the number of peptides released to the supernatant. Accordingly, Wang, et al. [31] found that extrusion may influence *N. oceanica* cell wall disruption and increase the release of valuable nutrients, such as polyunsaturated fatty acids and essential amino acids. The lack of results for the extraction of soluble proteins is possibly due to the high temperature applied during extrusion, which can alter the conformation of proteins leading to an irreversible protein unfolding and aggregation, as previously described for pigment–protein complexes in *A. platensis* [50,55].

Freeze-drying did not significantly influence protein extraction from *C. vulgaris* biomass. However, this pre-treatment was previously described as efficient in extracting proteins from algal biomass when combined with other pre-treatments. Unterlander, et al. [35] found that freeze-drying used as a pre-treatment for *C. vulgaris* biomass, followed by bead milling or sonication, improved soluble protein extraction up to 6-fold. This suggests that freeze-drying alone may not be sufficient to disrupt the cell wall and release nutrients.

Heating did not promote the release of proteins from *C. vulgaris* biomass, but, instead, it slightly but significantly increased the accumulation of protein fraction with 66 to 96 kDa in the pellet. The low temperature used in the present study (70 °C) may explain the lack of significant effect on protein solubility, although it caused the aggregation of a high-molecular-weight protein fraction in the pellet possibly as a result of protein denaturation and unfolding [50,55]. However, when higher (>70 °C) temperatures are applied to microalga biomass, protein denaturation and structural modification might occur less gradually, as shown for *A. platensis* [55], associated with the formation of complexes between lipids and other microalga compounds, which compromises the solvent extraction of nutrients. This aspect, together with the energy consumption required due to high drying temperatures, makes heating more difficult to scale up for industrial use and, ultimately, to be accepted as a potential treatment for microalgae [32]. Nevertheless, Abbassi, et al. [56] obtained a cell wall disruption of 94.6% when submitting *Nannochloropsis oculata* biomass to 40 °C and constant pressure (10 bars), with only an increment of about 4% when increasing the temperature up to 100 °C. Even considering the differences in the cell wall structure and composition between *N. oculata* and *C. vulgaris*, the previous results indicate that the effect of heating on *C. vulgaris* protein extraction deserves further exploitation.

## 5. Conclusions and Future Work

The results of this study indicate that bead milling, microwave, and sonication tended to insignificantly ($p > 0.05$) improve total protein content (7.43, 7.42, 7.33 mg/mL, respectively) compared to the control (6.66 mg/mL). In addition, bead milling and microwave caused a 1.2-fold increase in protein fraction with 32 to 40 kDa, whereas sonication decreased protein fraction with 26 kDa (1.39 to 1. 13 mg/mL) and increased the fraction of other proteins (2.73 to 3.88 mg/mL) compared to the control. Therefore, bead milling, microwave, and sonication are effective mechanical/physical pre-treatments for improving protein extraction from *C. vulgaris*. In contrast, extrusion, freeze-drying, and heating did not significantly affect protein denaturation and solubility under these experimental conditions. However, extrusion, which was not studied previously for *C. vulgaris*, resulted

in a threefold increase in total peptides released into the algal supernatant (32.4 μg/mL compared to 13.1 μg/mL in the control). In addition, this pre-treatment shows promise for industrial-scale applications, such as animal feeding. Therefore, our team is currently conducting in vivo trials with monogastric animals fed up to 15% of extruded *C. vulgaris* to determine the effect of this pre-treatment on the digestibility of microalgae nutrients.

Additional research is necessary to explore the effects of these pre-treatments, individually or with other mechanical/physical or enzymatic methods, on algal protein solubility and degradation. These findings have implications for the production of protein-rich feed supplements or ingredients from microalgae for livestock animals, and they provide a basis for future investigations into the optimization of protein extraction methods.

**Author Contributions:** Conceptualization, J.A.M.P.; data curation, M.P.S. and M.M.C.; formal analysis, M.P.S. and M.M.C.; investigation, M.P.S. and M.M.C.; writing—original draft preparation, M.P.S. and M.M.C.; writing—review and editing, J.A.M.P.; project administration, J.A.M.P.; funding acquisition, J.A.M.P. All authors have read and agreed to the published version of the manuscript.

**Funding:** This research was funded by Fundação para a Ciência e a Tecnologia grants (Lisbon, Portugal; UIDB/00276/2020 to CIISA, LA/P/0059/2020 to AL4AnimalS and a PhD grant UI/BD/153071/2022 to M.P.S.), and by Portugal2020 project (Lisbon, Portugal; P2020/17/SI/70114/2019 and associated researcher contract to M.M.C.).

**Institutional Review Board Statement:** Not applicable.

**Data Availability Statement:** The data presented in this study are available on request from the corresponding author.

**Conflicts of Interest:** The authors declare no conflict of interest.

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
