# Peer review of "Effect of Selected Mechanical/Physical Pre-Treatments on Chlorella vulgaris Protein Solubility"

_agriculture, doi:10.3390/agriculture13071309_

Round 1

Reviewer 1 Report

This paper describes the Effect of Selected Mechanical/Physical Pre-Treatments on Chlorella vulgaris Proteins Solubility.

They conclude that extrusion significantly increased the release of peptides from microalga biomass and, considering the industrial applicability, this method may be a good alternative for improving C. vulgaris protein bio-access.

The manuscript is clear, and presented in a structured format. The information enclosed in this work is valuable for scientist working in Chlorella.

The Introduction and the background are all right. The literature is well referenced and relevant. The manuscript structure follows the journal standards.

The conclusions are consistent with the arguments presented.

Mayor Corrections suggested:

Line 131-132: pre-treatment method should be described, not just the reference to other manuscripts. They are an essential part of this research.

Line133:

In Extrusion method for C. vulgaris is a bit different, since the temperature was 114 °C instead of 118

Line 136-137:

They should describe briefly the chemical composition determination methods

Fig 1 and 2: SDS-PAGE Molecular marker band weights are missing, they should be added.

Gel protein patterns in control proteins (no pretreated) from second supernatant electroforesis gel (Fig 2) are too different. These results diminish the reliability of the assay. These samples come from three different suspensions of Chlorella? Or the same? Please explain the possible reason of these differences

Line 242,243

They affirm that sonication was found to decrease the solubility of C. vulgaris proteins, leading to significant decreases in protein fractions with molecular weights of 66 to 96 kDa, 32 to 40 kDa and 26 kDa.

However, they only tested a single sonication condition (time and frequency). They should try other frequencies and times.

They should discuss the possible meaning of the higher bands in streets number 3 and 11 from Fig 1 second SDS-PAGE image.

I suggest to increase the confidence of the results by doing a western blott assay of your protein profiles with specific antibody for a housekeeping protein.

Author Response

Reviewer 1

This paper describes the Effect of Selected Mechanical/Physical Pre-Treatments on Chlorella vulgaris Proteins Solubility.

They conclude that extrusion significantly increased the release of peptides from microalga biomass and, considering the industrial applicability, this method may be a good alternative for improving C. vulgaris protein bio-access.

The manuscript is clear, and presented in a structured format. The information enclosed in this work is valuable for scientist working in Chlorella.

The Introduction and the background are all right. The literature is well referenced and relevant. The manuscript structure follows the journal standards.

The conclusions are consistent with the arguments presented.

Mayor Corrections suggested:

Reply: Thank you for your comments and suggestions. We appreciate it. We tried to address all of them.

Line 131-132: pre-treatment method should be described, not just the reference to other manuscripts. They are an essential part of this research.

Reply: Thank you for your comment and suggestion. Now a paragraph describing all the six pre-treatments is presented in section 2.1, lines 144-164, page 4.

Line133:

In Extrusion method for C. vulgaris is a bit different, since the temperature was 114 °C instead of 118

Reply: Thank you for your suggestion. The sentence was changed.

Line 136-137:

They should describe briefly the chemical composition determination methods

Reply: Thank you for your comment and suggestion. Now a brief section describing the chemical composition determination methods is presented in section 2.1, lines 165- 172, page 4

Fig 1 and 2: SDS-PAGE Molecular marker band weights are missing, they should be added.

Reply: Thank you for your suggestion. SDS-PAGE Molecular marker band weights were added in line 189, section 2.4, page 5; and also, in the captions of Figures 1 and 2, lines 246 and 284, pages 7 and 8, respectively.

Gel protein patterns in control proteins (no pretreated) from the second supernatant electrophoresis gel (Fig 2) are too different. These results diminish the reliability of the assay. These samples come from three different suspensions of Chlorella? Or the same? Please explain the possible reason of these differences

Reply: Thank you for your comment. Figure 2 presents images of SDS-PAGE gels of the pellet fraction showing the effect of pre-treatments on Chlorella vulgaris protein amount and solubility after 16 hours of incubation (n = 3). The three replicates came for the same sample and all the samples were previously homogenized before loading the gel. However, it is hard to load a completely homogeneous sample from pellet fractions in the gel due to considerable sample viscosity, which can explain the differences between gel protein patterns, despite the fact that the same person did all the SDS-PAGE gels to reduce errors.

Line 242,243

They affirm that sonication was found to decrease the solubility of C. vulgaris proteins, leading to significant decreases in protein fractions with molecular weights of 66 to 96 kDa, 32 to 40 kDa and 26 kDa.

However, they only tested a single sonication condition (time and frequency). They should try other frequencies and times.

Reply: Thank you for your comment. We appreciate it. We tested four different times for sonication (0, 15, 30 and 60 min), data not shown, and concluded that 7 cycles at 70% power for 15 min, with manual agitation in the middle time, was sufficient to partially disrupt C. vulgaris’ cell wall.

They should discuss the possible meaning of the higher bands in streets number 3 and 11 from Fig 1 second SDS-PAGE image.

Reply: Thank you for your comment. The pattern of C. vulgaris protein profile in SDS-PAGE gels is not fully characterized and understood. Streets number 3 and 11 correspond to a no pre-treatment sample supernatant and a bead milling sample supernatant. The same bands can be found in the other replicates of the same treatments but at lesser intensity due to our difficulties in homogenizing the samples. It is possible that these bands correspond to some high molecular weight proteins related to chloroplast or the cytoskeleton, as suggested by Tejano et al. 2019 (doi:10.3390/ijms20071786) and Piasecka et al. 2022 (doi: 10.3390/molecules27154817), and, thus, it would be interesting to characterize them in further research. I suggest increasing the confidence of the results by doing a western blott assay of your protein profiles with specific antibody for a housekeeping protein.

Reply: Thank you for your comment and suggestion. We appreciate it. We understand that doing a Western blot would be an improvement in our work. However, the aim of the present study was to compare the effect of the six pre-treatments on total protein quantification and protein fraction solubility of C. vulgaris’ proteins, and not to characterize the proteins’ profile and bioactive peptides from this microalga. We know that two important proteins from Chlorella vulgaris are located in the cytoskeleton (between 49 and 77 kDa) and chloroplast (between 70.9 and 80.7 kDa). Some pre-treatments can affect the release of a chloroplast-associated protein, rubisco, as explained in the reviewed paper, lines 303-307, page 9. Therefore, a Western blot using primary antibodies directed against Rubisco for chloroplast (Machida 2007, doi:10.1093/nass/nrm232; and Scherer 2019, doi:10.1016/j.algal.2019.101536) would be interesting to perform in a sequential study.

Further research to understand the full protein profile of C. vulgaris and its applications is reasonable and considered a good advance in our work.

Reviewer 2 Report

The manuscript has focused on the study of  different Pre-Treatment methods on Chlorella vulgaris Proteins Solubility . The results were reasonable. I would like to recommend the publication of this study if the following improvements could be made.

1)            It is difficult find a novelty in this study. How is it different from previous studies mentioned in the introduction section, and why does author think this study is necessary? At the end of the introduction, the aim of the work should be presented more clearly;

2)            The conclusions should contain information about the numerical values of the obtained main results

Author Response

Reviewer 2

The manuscript has focused on the study of different Pre-Treatment methods for Chlorella vulgaris Proteins Solubility. The results were reasonable. I would like to recommend the publication of this study if the following improvements could be made.

  • It is difficult to find a novelty in this study. How is it different from previous studies mentioned in the introduction section, and why does author think this study is necessary? At the end of the introduction, the aim of the work should be presented more clearly;

Reply: Thank you for your comment. Our aim was to evaluate and compare the six pre-treatments applied to C. vulgaris’ biomass in terms of their effectiveness, through cell wall degradation, on protein extraction and solubilization. Other studies compare one, two or a maximum of three methods, whereas the present study analyzed six different pre-treatments, one of them extrusion, which is a novelty as a pre-treatment applied to this microalga. Also, we analyzed the repercussions of the treatments on protein solubility and total protein content, which represents, and additional work compared to other reports. We reformulated the aim of the study, section 1, lines 127-136, page 3.

  • The conclusions should contain information about the numerical values of the obtained main results

Reply: Thank you for your comment and suggestion. We added numerical values of the main results obtained in the Conclusion, section 5, lines 385-393 and 399, page 10.

Reviewer 3 Report

This manuscript is about the comparison of different extraction techniques of proteins from microalgae, and I have some considerations:

- The techniques were not described, just indicated the references about it. Please include the description of methods and equipments

- Looking into references 37 and 38, these methods were used in different conditions turning difficult to compare the results. What is influencing the protein solubility? Temperature? Microwave energy? Abrasion in bead milling

In this way, I think this manuscript could be improved and reviewed in different papers adding control experiments to verify what it is important in extraction. Just now, it is very confusing

Author Response

Reviewer 3

This manuscript is about the comparison of different extraction techniques of proteins from microalgae, and I have some considerations:

- The techniques were not described, just indicated the references about it. Please include the description of methods and equipments

Reply: Thank you for your comment and suggestion. Now a paragraph describing all the six pre-treatments is presented in section 2.1, lines 144-164, page 4.

- Looking into references 37 and 38, these methods were used in different conditions turning difficult to compare the results. What is influencing the protein solubility? Temperature? Microwave energy? Abrasion in bead milling

Reply: Thank you for your comment. References 37 and 38 are previous papers concerning the effects of the same six pre-treatments combined or not with enzymes on protein solubility and total content, but in Arthrospira platensis.

Our objective is not to compare the results between microalgae. We want to compare each microalga, individually, the six pre-treatments in terms of total protein content and protein solubility.

We tested different methods of bead milling (homogenization in a shaker for 30 or 60 min), heating (70 or 90 °C, for 30 or 60 min, with microalga resuspension or just dried microalga) and microwave (microalga resuspension in keep warm method until it boiled or at 400W for 3 min). In the end, we selected the six methods using the same conditions for each microalga because we concluded that they were the best option (to avoid denaturation of microalgae’s proteins and compromise the nutritional composition).

-In this way, I think this manuscript could be improved and reviewed in different papers adding control experiments to verify what it is important in extraction. Just now, it is very confusing

Reply: Thank you for your comment. We appreciate it. Our aim with this work was to compare the six pre-treatments at the same time to evaluate which of them can be good methods to disrupt C. vulgaris’ rigid cell wall, with future application to in vivo studies in poultry nutrition. However, we agree that we should further evaluate all the six pre-treatments, individually, to clarify what variables are important for protein extraction by modifying the methods´ conditions, and, therefore, continue the first preliminary studies (data not shown) that we conducted as described for the previous comment.

Round 2

Reviewer 1 Report

The authors corrected all my suggestions. I recomend the publication of the article.

Author Response

Reviewer 1

The authors corrected all my suggestions. I recommend the publication of the article.

Reply: Thank you for your comments and suggestions. We appreciate it!

Reviewer 3 Report

No more comments

Author Response

Reviewer 3

No more comments.

Reply: Thank you for your comments and suggestions. We appreciate it!